# Effect of Silver Nanopowder on Mechanical, Thermal and Antimicrobial Properties of Kenaf/HDPE Composites

**DOI:** 10.3390/polym13223928

**Published:** 2021-11-14

**Authors:** Vikneswari Sanmuham, Mohamed Thariq Hameed Sultan, A. M. Radzi, Ahmad Adlie Shamsuri, Ain Umaira Md Shah, Syafiqah Nur Azrie Safri, Adi Azriff Basri

**Affiliations:** 1Department of Aerospace Engineering, Faculty of Engineering, Universiti Putra Malaysia, UPM Serdang 43400, Selangor Darul Ehsan, Malaysia; vikki.sanmuham@gmail.com (V.S.); adiazriff@upm.edu.my (A.A.B.); 2Laboratory of Biocomposite Technology, Institute of Tropical Forestry and Forest Products (INTROP), Universiti Putra Malaysia, UPM Serdang 43400, Selangor Darul Ehsan, Malaysia; adlie@upm.edu.my (A.A.S.); ainumaira91@gmail.com (A.U.M.S.); snasafri@gmail.com (S.N.A.S.); 3Aerospace Malaysia Innovation Centre (944751-A), Prime Minister’s Department, MIGHT Partnership Hub, Jalan Impact, Cyberjaya 63000, Selangor Darul Ehsan, Malaysia; 4Malaysia-Japan International Institute of Technology, Universiti Teknologi Malaysia, 54100 Kuala Lumpur, Malaysia

**Keywords:** silver nanopowder, kenaf, high-density polyethylene, antimicrobial

## Abstract

This study aims to investigate the effect of AgNPs on the mechanical, thermal and antimicrobial activity of kenaf/HDPE composites. AgNP material was prepared at different contents, from 0, 2, 4, 6, 8 to 10 wt%, by an internal mixer and hot compression at a temperature of 150 °C. Mechanical (tensile, modulus and elongation at break), thermal (TGA and DSC) and antimicrobial tests were performed to analyze behavior and inhibitory effects. The obtained results indicate that the effect of AgNP content displays improved tensile and modulus properties, as well as thermal and antimicrobial properties. The highest tensile stress is 5.07 MPa and was obtained at 10wt, TGA showed 10 wt% and had improved thermal stability and DSC showed improved stability with increased AgNP content. The findings of this study show the potential of incorporating AgNP concentrations as a secondary substitute to improve the performance in terms of mechanical, thermal and antimicrobial properties without treatment. The addition of AgNP content in polymer composite can be used as a secondary filler to improve the properties.

## 1. Introduction

Nowadays, nanotechnology is growing rapidly in the food and manufacturing industries for its effectiveness in terms of heat resistance and antimicrobial agents on products. The use of nanoparticles on composites brings advantages, especially in the aspect of dispersion. In addition, an increase in the nanoparticle content will reduce the hydrophobic properties and increase the mechanical, physical and thermal properties [1]. Due to their small size, these materials can interact with cellulose easily and disperse in composite systems. As written by Theodore et al. [2], the supermolecular structure of biopolymers contributes to the moisture absorption into cellulose, hemicelluloses and lignin. Nanoparticle usage in order to modify the surface of the lignocellulose-based composite can contribute to physical or chemical barriers to control the accumulation of moisture.

One of the nanomaterials used is silver nanopowders (AgNPs). AgNPs are increasingly used to produce various products, such as aerospace parts, medical, packaging and automotive because AgNPs have their uniqueness [3,4,5]. Due to the uniqueness of AgNP properties, various products have been produced, such as antibacterial agents, furniture, optical sensor health products, food industry products and anticancer agents [3,4,6]. AgNPs have the nano-sized potential to transform various physical and chemical properties of a substance and be used for a variety of purposes depending on an application [7]. Nurani et al. [8] declared that silver nanopowder as a secondary filler will be very promising since it has a high aspect ratio, which can determine surface-related properties such as solubility and stability. The high aspect ratio of silver nanoparticles exhibits microbial resistance and improves resistant strains, which are reliable to the current research concerns about optical, catalytic and microbial properties [8,9].

Natural fibers are known by industry and experts as elongated objects obtained from animals or plants. Natural fibers are cost-effective, have low-density properties, have low carbon sequestration, have good mechanical properties (high toughness and high specific strength) and are highly biodegradable [10,11,12,13]. Furthermore, natural fibers possess little manufacturing vitality in comparison to synthetic fibers, which are commonly used [10,14,15]. This would lead to producing polymer composites with very few production costs. Natural fibers can also help to improve environmental sustainability since they are eco-friendly composites [16,17,18]. The roughness and toughness of the fiber bundle will be arise the limitations in processing of natural fibers. Therefore, suitable chemical treatments have been proposed to overcome this problem [19]. The use of chemicals for surface treatment on natural fibers has various limitations, such as an expensive price, the surface fiber will shrink, and it requires high energy and processing time [20].

Kenaf (Hibiscus cannabinus L.) is an annual plant, which has historically been used for rope, twine, sackcloth and canvas. Kenaf fibers have many beneficial aspects, such as acceptable specific strength, economic viability, low density, reduced tool wear, cost-effectiveness, renewability, good biodegradability, reduced dermal and respiratory irritations, and energy savings by reducing the manufacturing process compared to synthetic fibers [21,22]. Based on Nishino et al. [23], it has been reported that kenaf composites have higher mechanical strength and thermal properties compared to other kinds of natural fiber polymer composites [24]. Kenaf is considered to be environmentally friendly due to several reasons, such as kenaf absorbs carbon dioxide more than any crop, no chemical pest control is needed, it removes toxic elements such as phosphorous from soil and it contributes to soil remediation [25]. Azammi et al. [26] and Yahya et al. [27] claimed that kenaf natural fiber-reinforced plastics are low weight and easy to fabricate; additionally, they are cost-effective and completely recyclable.

However, the research regarding the relationship between kenaf and silver-powder-reinforced HDPE and how to reduce surface treatment on fiber was never looked at. Apart from that, information regarding the combination of both materials is less. In this study, filler materials, such as organic or inorganic, in polymer systems can support main fillers in the strengthening of composites [28,29]. Therefore, the selection of AgNPs as a secondary filler was used to reduce fiber surface treatment [3] and improve the tensile, thermal and antimicrobial properties of kenaf/HDPE composites. The addition of silver nanopowder was suggested to improve the thermal stability of kenaf/HDPE composites. The antibacterial mechanism and silver nanopowder anti-biofilm may withstand the development of microorganisms and are expected to improve the sustainability of the kenaf/HDPE composites.

## 2. Materials and Methods

### 2.1. Materials

The main filler of the composites is kenaf with a size of 420 µm, which was obtained from the National Kenaf and Tobacco Board, Malaysia (NKTB). The secondary filler used in this research is a silver nanopowder (AgNP) with a size <100 nm. The matrix polymer that was used is High-Density Polyethylene (HDPE) with a density of 0.95 × 10^3^ kg/m^3^, which was purchased from Lotte Chemical Titan Sdn. Bhd. (Pasir Gudang, Malaysia). All materials in this study were used directly without any treatment or further refinement.

### 2.2. Sample Preparation

The kenaf and AgNPs were prepared through a Brabender Plastograph internal mixer as shown in Figure 1 to mix homogeneously at 150 °C for 50 min, and the speed was fixed at 60 rpm. HDPE with a mass of 24 g was added into the chamber, and then kenaf with a mass of 16 g [28] and AgNPs was varied to 0.82 g (2 wt%), 1.67 g (4 wt%), 2.55 g (6 wt%), 3.48 g (8 wt%), and 4.44 g (10 wt%). The composites were produced in the form of a sheet (1 mm) using a hot-press machine (Figure 1) at 150 °C and cooled for 5 min. All composites were dried in an oven for 24 h at a temperature of 70 °C. Figure 2 shows the samples before and after the compression process. All tests were carried out at Universiti Putra Malaysia (UPM), Malaysia.

### 2.3. Thermal Analysis

Thermal analysis is a technique for identifying the thermophilic and kinetic properties of materials. The technique used in this research is TGA (thermogravimetric analysis) and DSC (Differential Scanning Calorimetry). All tests were carried out at Universiti Putra Malaysia (UPM), Malaysia.

#### 2.3.1. Thermogravimetric Analysis (TGA) 

The thermal properties in composites are practically significant for studies related to temperature, and it is important to determine the degradation of kenaf/HDPE composites. Thermogravimetric analysis (TGA) was conducted with the TA Instrument TGA (Q500 model) (TA Instruments, New Castle, DE, USA) according to ASTM E1131 [30], which is displayed in Figure 3 for the ultimate decomposition temperatures of the prepared composite materials. All tests were carried out at Universiti Putra Malaysia (UPM), Malaysia.

The heating levels of 10 °C min^−1^ for the experiment were determined, with a temperature range of between 30 and 600 °C. Approximately 10 to 11 mg of sample was heated in a platinum sample pan of the prepared sample in a 50 mL min^−1^ flow rate of nitrogen gas atmosphere. All tests were carried out at Universiti Putra Malaysia (UPM), Malaysia.

#### 2.3.2. Differential Scanning Calorimetry (DSC)

A Differential Scanning Calorimeter (DSC) (TA Instruments, model DSC Q20, New Castle, DE, USA), as shown in Figure 4, was used to calculate the melting point temperatures of the blends that had been prepared. A DSC test was examined in a differential scanning calorimeter after heating the sample in the temperature range 35–200 °C at a heating rate of 10 °C min^−1^. All tests were carried out at Universiti Putra Malaysia (UPM), Malaysia.

### 2.4. Antimicrobial Analysis

Two kinds of mediums were utilized: bacteria-growing sterile Mueller–Hinton agar (MHA) and yeast-growing sterile polydopamine (PDA). All tests were carried out at Universiti Putra Malaysia (UPM), Malaysia.

By incubating pure micro-organism culture at 37 °C, for 18 h, an inoculum suspension was created. The implanted microbes were shifted into the liquid media to 0.5 McFarland turbidity standard for adjusting their turbidity. With a sterile cotton swab, a sample of Escherichia coli (E. coli), UPMC 25921, Pseudomonas aeruginosa (P. aeruginosa) ATCC 15442, and Candida albicans (*C. albicans*) ATCC 90029 was dispersed over the entire agar surface and then swabbed over the sampling area. The kenaf/HDPE composites with and without AgNP samples were cut into 6 mm discs in diameter prior to testing and sterilized for 30 min using UV radiation. Then, put it into the Petri platter. A Petri dish was incubated for 18 to 24 h at 37 °C for bacteria, but for fungi it was incubated for 72 h at 30 °C. A clear area has been observed and the inhibition area size has been registered. Every sample was performed three times.

## 3. Results and Discussion

### 3.1. Tensile Properties

The tensile stress properties of kenaf/HDPE biocomposites with five different concentrations of AgNPs, which are 0%, 2%, 4%, 6%, 8% and 10%, were demonstrated. The changes in the failure of each composite tested can be seen in its properties. The results of the test determine whether the composite is acceptable or not in terms of function, structure and visuals. Figure 5 and Figure 6 show the tensile stress and modulus properties of kenaf/HDPE composites with the different AgNP contents. A summary of the analysis of mechanical properties is also presented. From the result, the trends of tensile stress and modulus show an increase with increasing AgNP contents compare to without AgNPs. This indicates exceptional compatibility between kenaf/HDPE composites shows a major increase in their tensile stress at the values of 3.78, 4.03, 4.15, 4.67 and 5.07 MPa with the addition of AgNPs of 0 to 10%, respectively. The main factor of this increase is due to the physical interaction between the components. It has been proven by Rhim [4] that the interaction and combination of materials need to have a relationship between all the components to ensure increased tensile properties. Besides, the performance increase is also due to the nature of AgNPs having a nano-chain structure and high nano-mechanical performance [31].

Due to the very limited inclusion of AgNPs in the kenaf/HDPE composites, slight increases in the tensile stress and tensile modulus were predicted. Figure 6 shows the tensile modulus of the kenaf/HDPE composites with different AgNP contents. The figure shows a similar trend with tensile stress increased with increasing the AgNP content. The increase in tensile modulus is due to several factors, such as good adhesion, abrasive surface, less void and microcracking [32] between kenaf and AgNP fillers. Moreover, AgNPs could act as a secondary filler in kenaf/HDPE composite to enhance the tensile and stiffness characteristics. N. Sapiai et al. [33] reported that the efficiency of a filler also depends on various factors, such as filler dispersion, filler volume, the type of matrix used and matrix bonding [34]. Such a relationship was also reported for composites such as kenaf and dolomite-reinforced Low-Density Polyethylene (LDPE) composites [28]. Therefore, dolomite in the composite material could act as a secondary filler to enhance their stiffness characteristics.

Figure 7 shows the elongation at the break of the kenaf/HDPE composites with different concentrations of AgNPs. From the observations, the elongation of kenaf/HDPE composites shows decreased with increasing the AgNP content compared to composites without AgNPs. This is due to the higher ductile behavior of kenaf/HDPE composites with increased AgNP content compared to those without AgNPs. The results for the elongation at break for various percentages of AgNP addition in kenaf/HDPE composites are illustrated in Figure 7. Such results show the decrease in elongation as a larger percentage of AgNPs is added to the composites. The kenaf/HDPE composites with a maximum addition of 10% AgNPs were able to display the lowest elongation, among other samples. The elongation at break follows a reverse trend with that of tensile strength and modulus. As the concentration of AgNPs increases, the amount of elongation at break decreased. This phenomenon may be due to excessive polymer and fiber interactions, hard domain rotation, interface slippages and fibrillation on the matrix [35,36]. In addition, it might occur because AgNP content is a large percentage of crystallinity, slowly losing its ductile properties, with additional rigid silver nanoparticles. The impact of AgNP addition to kenaf/HDPE composites has, therefore, been shown to be more essential for tensile stress and tensile modulus, whereas the elongation is dropped significantly due to the addition of AgNPs that restrict the molecule chain to move, which makes the chain lose its ductility and high in stiffness.

### 3.2. Thermal Properties

#### 3.2.1. Thermogravimetric Analysis (TGA)

TGA measured the thermal stability of the pure kenaf/HDPE composites with different concentrations of silver nanopowder. The degradation temperatures of kenaf/HDPE composites with different contents of AgNPs (0 %,2 %,4 %,6 %,8 % and 10 %) are exhibited in the TGA graph shown in Figure 8. The curves indicate that thermal degradation only occurred after some concentrations of heat energy had been incorporated into the components. Additionally, it is clear that all the kenaf/HDPE composites with different AgNP contents showed that there was no significant difference in the value of thermal stability from each other. The onset degradation temperature was observed.

The experimental result shows the same thermal degradation trend for all composites. The trend graph shows a decrease for all composites with and without AgNP content. It is also observed from the diagram that kenaf/HDPE composites with AgNP content are more stable compared to those without AgNPs at a temperature of 400–550 °C. In thermal experiments, there are four visible phases. The first phase is at a temperature of 30–100 °C, where, at this temperature, the mass of the sample will decrease because the evaporation moisture process takes place on all composite samples. Due to naturally hydrophilic characteristics, kenaf has moisture content at room temperature and humidity before the composite fabrication process [32,37,38]. The second phase is the decomposition of the element lignocellulosic at a temperature range of 200–320 °C [13,39], followed by the third phase, which is the decomposition of hemicellulose at a temperature range of 350–450 °C [40,41], and finally, the last phase is the decomposition of cellulose lignin and ash [16]. Table 1 shows the result degradation of different AgNP contents on kenaf/HDPE composites. From the result, AgNP wt10% shows that the highest percentage char residue was 12.96%.

The result shows that the addition of AgNP content of kenaf/HDPE composites showed an increase in thermal stability compared to without AgNPs. It has been known that AgNPs have excellent thermal stability and volume ratio properties [3,42]. With the increase in the AgNP content, the thermal decomposition properties of kenaf/HDPE composites were further increased. This occurs due to the increment of the molecular weight of the kenaf/HDPE composites. Besides, the addition of AgNP content of kenaf/HDPE composites has improved its thermal properties and also because the AgNP content has high decomposition properties. The same finding has been investigated by [28], where the addition of other materials, such as dolomite, as well as increasing its content into the composite system can improve the thermal stability properties. The TGA thermograms are of the LDPE/kenaf composites with different contents of dolomite, and the dolomite possesses the highest thermal decomposition characteristics. With the increase in the content of dolomite in the composites, the thermal stability properties also increase, and this is due to the properties of dolomite possessing the highest thermal decomposition characteristics. Overall, the addition of AgNP content as a secondary filler provides benefits to improve the thermal stability properties of kenaf/HDPE composites.

#### 3.2.2. Differential Scanning Calorimetry (DSC)

The DSC thermograms, melting point (*T*_m_) and glass transition temperature (*T*_g_) values of the kenaf/HDPE composites with and without AgNPs are shown in Figure 9 and Table 2.

DSC study on materials is important because it can analyze the properties of composite materials [43]. From Figure 9, the glass transition temperatures, *T*_m_ (midpoint values), of kenaf/HDPE composites with AgNPs are greater than those of kenaf/HDPE composites without AgNPs. The results indicate that the addition of AgNPs tends to increase the prohibition of the movement of the polymer matrix chain. The *T*_m_ (onset value) of the kenaf/HDPE composites also significantly increased with increasing the AgNPs contents from 2 wt% to 10 wt%, and it indicates that the addition of AgNPs into Kenaf/HDPE composites is more stable than without AgNP content. Kenaf/HDPE composites with more AgNPs have a *T*_m_ (onset value) slightly higher than those with less concentration of AgNPs, and therefore the thermal properties of kenaf/HDPE composites have also been impacted by AgNPs. The *T*_m_ (onset value) for kenaf/HDPE with 10 wt% is 128.53 °C when compared to those with a lower concentration of AgNPs. The *T*_m_ (onset value) increases due to AgNP content, which can make the structure between kenaf and HDPE change and increase the structure by the content of AgNPs. Combining AgNPs into composites enhanced the thermal stability and was similarly reported by another author [44,45,46]. Variations in the *T*_m_ (midpoint value) might also adversely determine the compatibility of the samples. From Table 2, Tm (midpoint value) levels of kenaf/HDPE composites with 0 wt% of AgNPs is lesser (129.97 °C) than kenaf/HDPE composites with 10 wt% of AgNPs (133.83 °C). The intermolecular forces between kenaf/HDPE composites matrixes are due to AgNPs in mixes, which can strengthen the interfacial bond between mixing components [37,45]. The combination of kenaf/HDPE composites with AgNPs provided the advantages to produce quality multifunctional materials and it is also suitable for various manufacturing applications.

### 3.3. Antimicrobial Testing

The test is usually conducted for the antibiotic sample that is most successful in treating bacteria or fungal infections. Antimicrobial/Antifungal Test Microbial reactions to antimicrobial agents distinguish from each other (sensitivities/resistances). Two widespread pathogenic bacterial, P. aeruginosa and E. coli, were chosen to assess the antibacterial activity of the kenaf/HDPE with Silver Nanopowder (AgNP), while *C. albicans* was analyzed for the fungal inhibition in Table 3. The bacteria that is commonly used in research can be sorted into two groups, i.e., Gram-negative bacteria and Gram-positive bacteria. In this research, foodborne pathogenic bacteria, which are E. *coli* (ATCC 25922) and *P. aeruginosa* (ATCC 15442) [47], were used to carry out the antibacterial testing.

Based on Table 3 and Table 4, the kenaf/HDPE composites with AgNPs revealed a stronger activity against the gram-negative bacteria compared without AgNP composition, which is significantly lower than the activity seen in the earlier literature [5,48]. Antimicrobial variation may be attributable to their different configurations of cell walls [42,49]. According to Priyadarshini et.al. [50], Gram-negative bacteria consist of thin peptidoglycan (7–8 nm). Consequently, the thin layer of peptidoglycan allows the penetration of silver nanoparticles in microbial cells to destroy. Additional microorganisms, specifically the pathogenic fungus *C. albicans*, were also tested, and the processed kenaf/HDPE composites filled with AgNPs exhibited greater areas of inhibition compared to Gram-negative bacteria, which followed the work of Veranitisagul et al. [49]. Once again, the process is specifically connected to the microorganisms’ cell wall structure.

Kenaf/HDPE composites with 10wt% of AgNPs demonstrated the strongest inhibition zone against E. coli (ATCC 25922) above all the samples. *C. albicans*, underlining the significance of Ag particles in 8wt% and 10wt% of kenaf/HDPE composites, filled AgNP contents. The findings show that silver nanoparticles exhibit greater sensitivity towards Pseudomonas aeruginosa than Escherichia coli in extracts for bacterial activity. Such bacteria displayed an inhibition zone ranging from 10–15 to 10–13 mm, respectively. 

Table 5 shows the inhibition zone measurement of bacteria, whereas Table 6 shows the inhibition zone measurement of yeast that had been used for the antimicrobial testing.

Table 6 displays improved results for fungal infection relative to the Candida albicans test where positive results are acquired for samples of kenaf/HDPE composites with 8 wt% and 10 wt % of AgNPs. Silver nanopowder and the microbial reaction were specifically on the surface when the materials were placed for a long time in the surrounding environment. This stresses the fact that although kenaf/HDPE composites filled AgNPs, they are antimicrobial and biodegradable materials. Those kenaf/HDPE-composite-filled AgNPs may also be one of the potential technologies for forthcoming plastics that are environmentally sustainable.

## 4. Conclusions

The effect of AgNPs contents on tensile, thermal and antimicrobial properties of kenaf/HDPE composites was investigated in this study. The tensile strength and modulus of the kenaf/HDPE composites show improvement with the presence of AgNPs, and elongation at break was reduced at the highest content, as shown by the mechanical testing result. Kenaf/HDPE composites revealed the highest tensile stress and tensile modulus at 10 wt% AgNPs, which are 5.05 MPa and 5.23 GPa, respectively. Meanwhile, with regard to thermal properties, there was a large increase in the alternative values of *T*_g_ and *T*_m_ due to the participation of AgNP content. The temperature of *T*_g_ has changed positively. Relative to kenaf/HDPE composites, HDPE with 10 wt% AgNP content displayed the maximum degradation temperature of 496 °C. The main factors in this research are kenaf/HDPE composites with 10 wt% of AgNPs that exhibit antimicrobial activity against E. coli, P. aeruginosa and *C. albicans* inhibition using an agar disk diffusion test as an output. Kenaf/HDPE composites with AgNP content are expected to be one of the reasonable manufacturing materials, as they showed strong antimicrobial properties that may minimize the requirements for the cleaning process. The findings obtained include the fact that AgNPs can inhibit the growth of microbes. This shows that kenaf/HDPE composites are biodegradable and exhibit antimicrobial properties that can be implemented in the interior part of aircraft and food packaging.

## Figures and Tables

**Figure 1 polymers-13-03928-f001:**
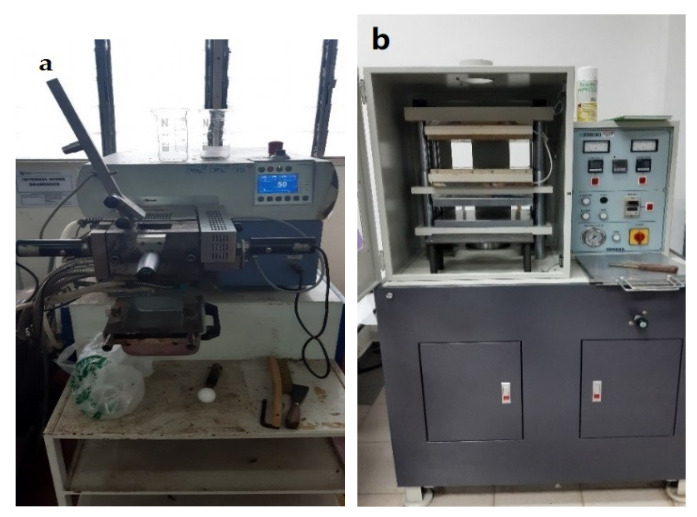
(**a**) Brabender Plastograph internal and (**b**) hot press machine.

**Figure 2 polymers-13-03928-f002:**
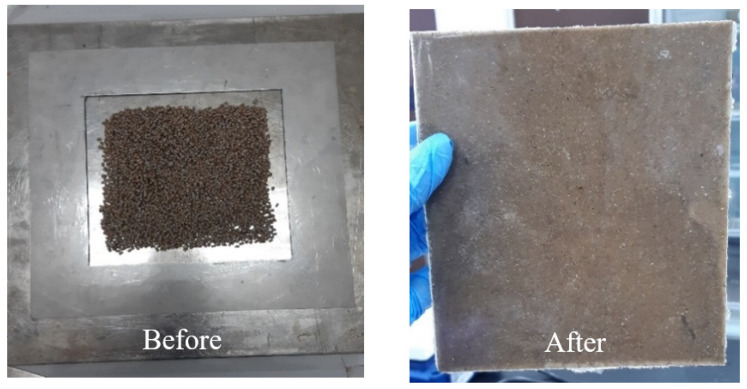
Sample before and after the compression process.

**Figure 3 polymers-13-03928-f003:**
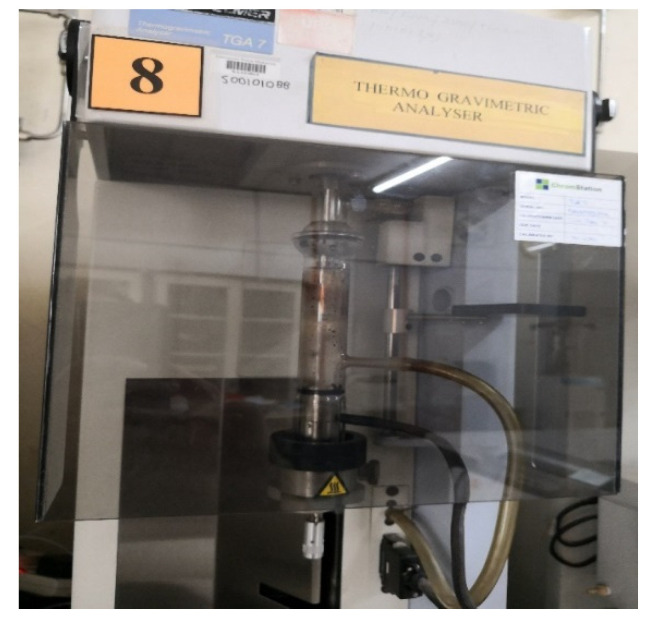
Thermogravimetric analyzer.

**Figure 4 polymers-13-03928-f004:**
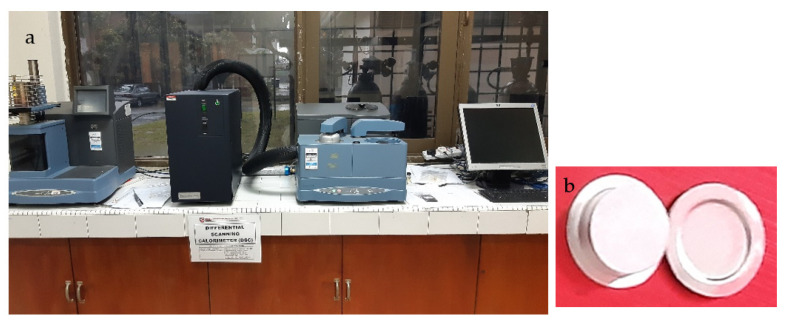
(**a**) Differential Scanning Calorimeter (DSC) and (**b**) aluminum sample pan (5 mm).

**Figure 5 polymers-13-03928-f005:**
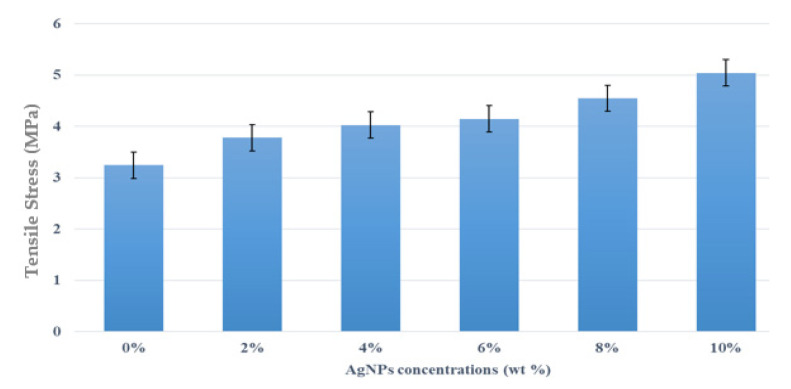
Effects of AgNP content on tensile stress of kenaf/HDPE composites.

**Figure 6 polymers-13-03928-f006:**
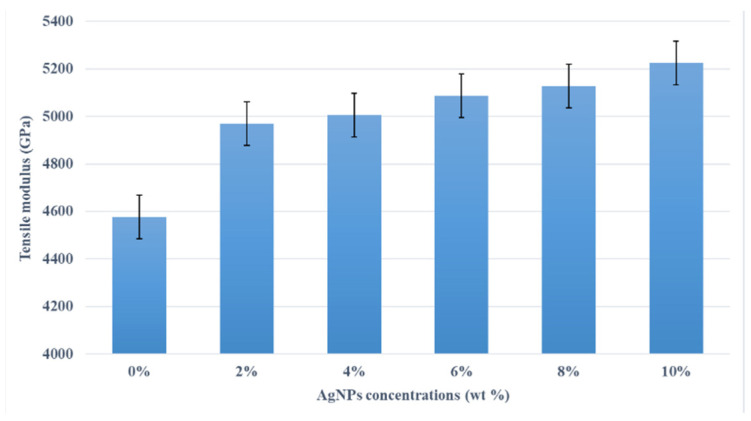
Effects of AgNP content on tensile modulus of kenaf/HDPE composites.

**Figure 7 polymers-13-03928-f007:**
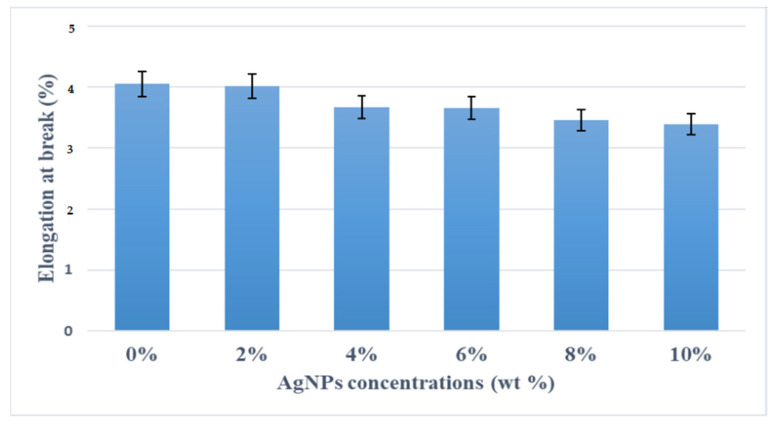
Effects of AgNP content on elongation at break of kenaf/HDPE composites.

**Figure 8 polymers-13-03928-f008:**
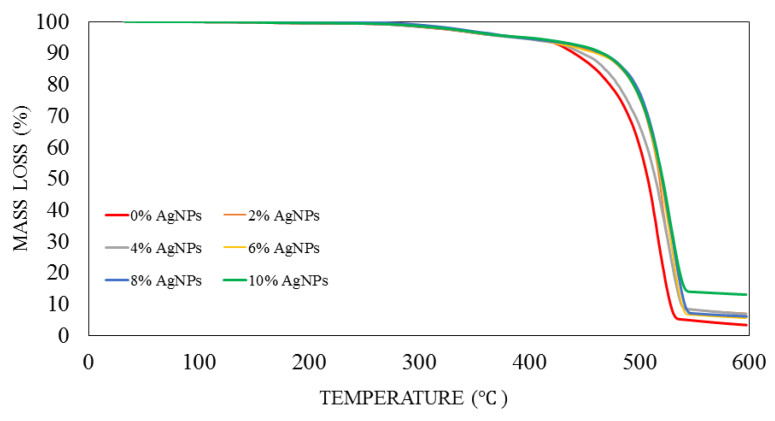
TGA of kenaf/HDPE composites at different AgNP contents.

**Figure 9 polymers-13-03928-f009:**
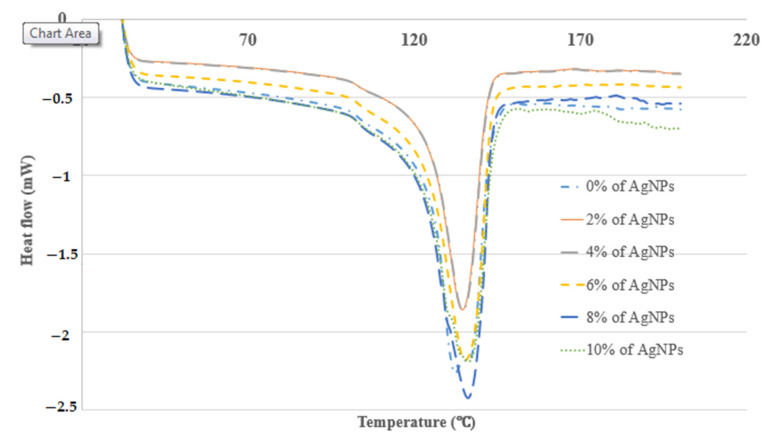
DSC thermograms of the kenaf/HDPE composites containing AgNPs.

**Table 1 polymers-13-03928-t001:** The initial and maximum degradation temperature values of the kenaf/HDPE composites from TGA thermograms.

AgNPs (%)	Maximum Degradation Temperature (℃)	Weight Loss (%)	Char at 600 °C (wt%)
0	472	80.45	3.27
2	479	80.42	6.02
4	493	80.40	6.97
6	494	80.43	5.47
8	495	80.50	6.77
10	496	80.41	12.96

**Table 2 polymers-13-03928-t002:** Thermal properties of kenaf/HDPE composites without and with AgNPs, as measured from DSC thermograms.

AgNPs (wt %)	*T*_m_ (Onset Value) (°C)	*T*_m_ (Midpoint Value) (°C)
0	120.76	129.97
2	120.90	129.98
4	121.17	132.20
6	121.90	132.75
8	122.78	133.45
10	128.53	133.83

**Table 3 polymers-13-03928-t003:** Antimicrobial properties of kenaf/HDPE composite materials.

Concentration of AgNPs (wt %)	Target Microbes		
	Escherichia Coli UPMC 25922	Pseudomonas Aeruginosa ATCC 15442	Candida Albicans ATCC 90028
0	−	−	−
2	−	−	−
4	−	−	−
6	−	−	−
8	−	−	+
10	+	+	+

+ve: the presence of inhibition zone against microbes. −ve: no indication of inhibition zone against microbes.

**Table 4 polymers-13-03928-t004:** Antibacterial study of kenaf/HDPE composites.

Microbes Concentration of AgNPs (wt%)	Inhibition Zone (mm)	
	Escherichia Coli UPMC 25922	Pseudomonas Aeruginosa ATCC 15442
10	12.2	14.3

**Table 5 polymers-13-03928-t005:** Antifungal study of kenaf/HDPE composites.

Microbes Concentration of AgNPs (wt%)	Inhibition Zone (mm)
	Candida Albicans ATCC 90028
8	6
10	17.5

**Table 6 polymers-13-03928-t006:** Positive tests for antimicrobial properties on E. coli, P. aeruginosa and *C. albicans* using the agar disk diffusion method.

Microbes	Images
Escherichia coli UPMC 25922	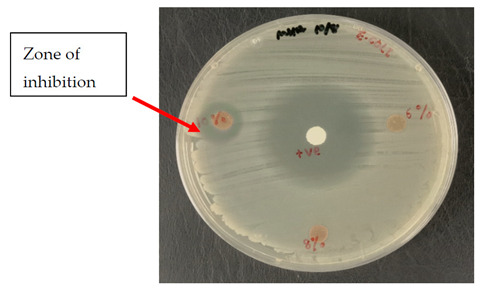
Pseudomonas aeruginosa ATCC 15442	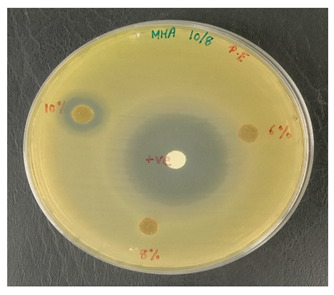
Candida albicans ATCC 90028	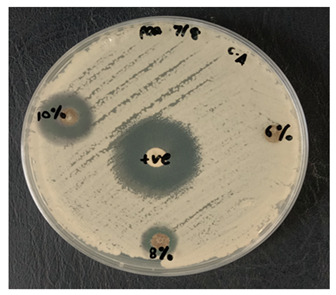

## Data Availability

The data presented in this study are available on request from the corresponding author.

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
