# Peer review of "Effect of Silver Nanopowder on Mechanical, Thermal and Antimicrobial Properties of Kenaf/HDPE Composites"

_polymers, 2021, doi:10.3390/polym13223928_

Round 1

Reviewer 1 Report

The manuscript describes the effect of adding silver nanopowder to kenaf/HDPE composites with respect to mechanical, thermal and anti-microbial properties. The manuscript lacks several aspects of a good publication. Some points to consider are given below: 1. Novelty of the work is not explained anywhere in the manuscript. The versatility of this work with similar works are not all considered by the authors. 2. The data presented in the manuscript are suspicious and many superlative claims are included with respect to them. For example, the data of mechanical properties. 3. Abstract: dan inhibitory effects: what is this? 4. Line 30: sentence is not complete 5. Figure 6: Y-axis label and the figure caption are not matching. Also the values 6. Figure 7: Check the values and unit. 7. Figure 8: Even though the trend is decreasing for elongation at break, the numerical values are in the error range. Better not to claim much on these values. 8. Line 263: modulus reduction: is it so? 9. Line 263: polymer-fill interaction: is it filler? 10. Line 254: APGNs 11. Line 276: Graph curves: either one of graph or curves can be deleted. 12. Line 279: How can you measure thermal conductivity from TGA? 13. AGNPs and AGNps- be consistent on one notation. 14. Capitalization of the first word of sentences is missing in several occasions. 15. Tables 1 and 2: The digits after decimal points are two or three which means that the instruments are so sensitive in the measurements. As far as I know, both these machines are not so sensitive to report such values. Better to avoid the decimal points and convert them to digits only. 16. Line 339: How DSC determine the strength of the material? 17. Tg values are given in the range of 120-128 degree Celsius for the composites. Are you sure about these values? The matrix used here is HDPE and the values are in the range of 0 to -10 or so. Some serious mistake happened here.

Author Response

The manuscript describes the effect of adding silver nanopowder to kenaf/HDPE composites with respect to mechanical, thermal and anti-microbial properties. The manuscript lacks several aspects of a good publication. Some points to consider are given below:

  1. Novelty of the work is not explained anywhere in the manuscript. The versatility of this work with similar works are not all considered by the authors.

Thank you for your kind suggestion. All necessary changes has been made as per suggestion.

  1. The data presented in the manuscript are suspicious and many superlative claims are included with respect to them. For example, the data of mechanical properties.

The error bars of the results of mechanical properties have been added, we hope it will dispel the doubt of the reviewer.

  1. Abstract: dan inhibitory effects: what is this?

The word should be “and”. Thank you for your kind suggestion. All necessary changes has been made as per suggestion.

  1. Line 30: sentence is not complete

Thank you for your kind suggestion. All necessary changes has been made as per suggestion.

  1. Figure 6: Y-axis label and the figure caption are not matching. Also the values

Thank you for your kind suggestion. All necessary changes has been made as per suggestion

  1. Figure 7: Check the values and unit.

Thank you for your kind suggestion. All necessary changes has been made as per suggestion

  1. Figure 8: Even though the trend is decreasing for elongation at break, the numerical values are in the error range. Better not to claim much on these values.

Thank you for your kind suggestion. All necessary changes has been made as per suggestion

  1. Line 263: modulus reduction: is it so?

Thank you for your kind suggestion. All necessary changes has been made as per suggestion

  1. Line 263: polymer-fill interaction: is it filler?

The word is “fibres”. Thank you for your kind suggestion. All necessary changes has been made as per suggestion

  1. Line 254: APGNs

Thank you for your kind suggestion. All necessary changes has been made as per suggestion

  1. Line 276: Graph curves: either one of graph or curves can be deleted.

Thank you for your kind suggestion. All necessary changes has been made as per suggestion

  1. Line 279: How can you measure thermal conductivity from TGA?

The term has been changed. Thank you for noticing the mistake. All necessary changes has been made as per suggestion

  1. AGNPs and AGNps- be consistent on one notation.

Thank you for your kind suggestion. All necessary changes has been made as per suggestion

  1. Capitalization of the first word of sentences is missing in several occasions.

Thank you for your kind suggestion. All necessary changes has been made as per suggestion

  1. Tables 1 and 2: The digits after decimal points are two or three which means that the instruments are so sensitive in the measurements. As far as I know, both these machines are not so sensitive to report such values. Better to avoid the decimal points and convert them to digits only.

Table 1: decimal points have been removed

Table 2: need to show decimal points to show the difference. If rounded, the values are almost the same

The sentence is removed. Thank you for your kind suggestion. All necessary changes have been made as per suggestion

  1. Line 339: How DSC determine the strength of the material?

The sentence is removed. Thank you for your kind suggestion. All necessary changes have been made as per suggestion

  1. Tg values are given in the range of 120-128 degree Celsius for the composites. Are you sure about these values? The matrix used here is HDPE and the values are in the range of 0 to -10 or so. Some serious mistake happened here.

The misinterpreted values have been corrected. Thank you for raising this matter. All necessary changes have been made as per suggestion

Reviewer 2 Report

Overall, this paper present a good finding that is worth for publication, however, some improvement is necessary before this paper is ready to be published.

Abstract:

  • Some sentence from the template were found in the first line of abstract.
  • The abstract should cover Introduction, Methodology, Results, and overall conclusion. Please make sure all this item is in place.
  • There is no significant results mentioned in the abstract. The author should highlight the improvement in the tensile strength/modulus, improvement in the thermal stability etc instead of generally describing that these properties were improved without any supporting information.
  • For thermal, there is no specific results for TGA and DMTA which indicates the improvement of thermal stability.
  • Some sentence need to be clarify in terms of the structure to give more clear information to the reader.
  •  

Introduction:

  • The introduction cover most aspect of this study, which is good, however;
  • Some of the sentence is confusing and requires proof reading from professional English proof reading service.
  • At the end of introduction, the end of the literature should be emphasize with research gap in the literature in order to show the importance of this research to be carried out. This can be followed by objective of study, the details of research and expected results should not be mentioned here.

Methodology:

  • Please mentioned the name the country for all equipment use in this study (processing and characterization).
  • This section need to be written in passive voice and requires professional English proof reading service.
  • Figure 5 is not appropriate to be included in this manuscript.
  • SEM is an important characterization for composites especially in the tensile fracture, however, this paper does not provide this results.
  •  

Results and Discussion

  • There are mistake in Figure 6 y-axis. (should be tensile strength right?) please check.
  • Why the author stops the inclusion of AgNP at 10%? Since the results keep on showing improvement on the mechanical properties and why did the author choose 2,4,6,8, and 10% AgNP as the amount to be used in this study? Any justification.

  • For the mechanical properties, the author should mentioned the general trend of the composites behavior first before go deep into the discussion. Author should explain why AgNP can increase the strength of the composite, this should be supported with appropriate argument and citation of previous studies.
  • For DSC, why the author discuss the results by explaining the decrement of Tg/Tm while the AgNP decrease? The discussion should focus on the effect of increasing AgNP on the Tg and Tm behavior. Please revise. In addition, please provide strong justification on why increasing AgNP helps to increase the Tg and Tm of the composites, and supported by appropriate argument with reference.

Conclusion

  • This sentence “The lowest decomposition temperature is 522.5 °C.” is referred to what sample? Please complete the sentence since in conclusion, one sentence might stand alone and not related to the previous one.
  • There is no need to provide justification on the improvement of properties in the conclusion, please check so that the conclusion became more concise and accurate.
  • The word “so on” is not appropriate here.

Overall comment, this paper requires professional English proof reading prior to publication.

Author Response

  1. Some sentence from the template were found in the first line of abstract.

Thank you for your kind suggestion. All necessary changes has been made as per suggestion.

  1. The abstract should cover Introduction, Methodology, Results, and overall conclusion. Please make sure all this item is in place.

Thank you for your kind suggestion. All necessary changes has been made as per suggestion.

  1. There is no significant results mentioned in the abstract. The author should highlight the improvement in the tensile strength/modulus, improvement in the thermal stability etc instead of generally describing that these properties were improved without any supporting information.

Thank you for your kind suggestion. All necessary changes has been made as per suggestion.

  1. For thermal, there is no specific results for TGA and DMTA which indicates the improvement of thermal stability.

Thank you for your kind suggestion. All necessary changes has been made as per suggestion

  1. Some sentence need to be clarify in terms of the structure to give more clear information to the reader.

Thank you for your kind suggestion. All necessary changes has been made as per suggestion

Introduction:

  1. The introduction cover most aspect of this study, which is good, however;

Thank you for your kind suggestion. All necessary changes has been made as per suggestion

  1. Some of the sentence is confusing and requires proof reading from professional English proof reading service.

Before submitting this manuscript to the journal, we have checked it using grammar and

spell checker software. Additionally, we do not have time to send this manuscript to a

company which provides English proofreading and editing services because the time

taken is more than 10 days. It will surpass the revision due date.

  1. At the end of introduction, the end of the literature should be emphasize with research gap in the literature in order to show the importance of this research to be carried out. This can be followed by objective of study, the details of research and expected results should not be mentioned here.

Thank you for your kind suggestion. All necessary changes has been made as per suggestion

Methodology:

  1. Please mentioned the name the country for all equipment use in this study (processing and characterization).

Thank you for your kind suggestion. All necessary changes has been made as per suggestion

  1. This section need to be written in passive voice and requires professional English proof reading service.

Thank you for your kind suggestion. All necessary changes has been made as per suggestion

  1. Figure 5 is not appropriate to be included in this manuscript.

Thank you for your kind suggestion. All necessary changes has been made as per suggestion

  1. SEM is an important characterization for composites especially in the tensile fracture, however, this paper does not provide this results.

We apologize for no morphology due to constraints to enter the lab during the Covid 19 pandemic. In the future, we will make it. Thank you for your kind suggestion. All necessary changes has been made as per suggestion

Results and Discussion

  1. There are mistake in Figure 6 y-axis. (should be tensile strength right?) please check. Tensile stress. Thank you for your kind suggestion. All necessary changes has been made as per suggestion

  1. Why the author stops the inclusion of AgNP at 10%? Since the results keep on showing improvement on the mechanical properties and why did the author choose 2,4,6,8, and 10% AgNP as the amount to be used in this study? Any justification.

Based on the previous studies, we have found that 10% of nanopowder is the maximum

content which usually added into the composites. In addition, we only have a limited

source of AgNP, which hindered us from exploring more.

  1. For the mechanical properties, the author should mentioned the general trend of the composites behavior first before go deep into the discussion. Author should explain why AgNP can increase the strength of the composite, this should be supported with appropriate argument and citation of previous studies.

Thank you for your kind suggestion. All necessary changes has been made as per suggestion

  1. For DSC, why the author discuss the results by explaining the decrement of Tg/Tm while the AgNP decrease? The discussion should focus on the effect of increasing AgNP on the Tg and Tm behavior. Please revise. In addition, please provide strong justification on why increasing AgNP helps to increase the Tg and Tm of the composites, and supported by appropriate argument with reference.

Thank you for your kind suggestion. All necessary changes has been made as per suggestion

   Conclusion

  1. This sentence “The lowest decomposition temperature is 522.5 °C.” is referred to what sample? Please complete the sentence since in conclusion, one sentence might stand alone and not related to the previous one.

Removed this sentence. Thank you for your kind suggestion. All necessary changes has been made as per suggestion

  1. There is no need to provide justification on the improvement of properties in the conclusion, please check so that the conclusion became more concise and accurate.

Removed this sentence. Thank you for your kind suggestion. All necessary changes has been made as per suggestion

  1. The word “so on” is not appropriate here.

Removed this sentence. Thank you for your kind suggestion. All necessary changes has been made as per suggestion

  1. Overall comment, this paper requires professional English proof reading prior to publication.

If this manuscript is accepted for publication, we would send it to the company that provides English proofreading and editing services. Thank you for your kind recommendation.